# Slow-light-enhanced energy efficiency for graphene microheaters on silicon photonic crystal waveguides

Siqi Yan[1,2], Xiaolong Zhu[2,3], Lars Hagedorn Frandsen[2], Sanshui Xiao[2,3], N. Asger Mortensen[2,3], Jianji Dong[1] & Yunhong Ding[2]

Slow light has been widely utilized to obtain enhanced nonlinearities, enhanced spontaneous emissions and increased phase shifts owing to its ability to promote light–matter interactions. By incorporating a graphene on a slow-light silicon photonic crystal waveguide, here we experimentally demonstrate an energy-efficient graphene microheater with a tuning efficiency of $1.07 \, \text{nm} \, \text{mW}^{-1}$ and power consumption per free spectral range of 3.99 mW. The rise and decay times (10–90%) are only 750 and 525 ns, which, to the best of our knowledge, are the fastest reported response times for microheaters in silicon photonics. The corresponding figure of merit of the device is 2.543 nW s, one order of magnitude better than results reported in previous studies. The influence of the length and shape of the graphene heater to the tuning efficiency is further investigated, providing valuable guidelines for enhancing the tuning efficiency of the graphene microheater.

[1] Wuhan National Laboratory for Optoelectronics, Huazhong University of Science and Technology, 430074 Wuhan, China. [2] Department of Photonics Engineering, Technical University of Denmark, DK-2800 Kongens Lyngby, Denmark. [3] Center for Nanostructured Graphene, Technical University of Denmark, DK-2800 Kongens Lyngby, Denmark. Correspondence and requests for materials should be addressed to J.D. (email: jjdong@mail.hust.edu.cn) or to Y.D. (email: yudin@fotonik.dtu.dk).

Following several decades of explosive growth, the information industry has become a major energy consumer owing to the high power consumption by data centres[1]. As one of the most promising candidates for satisfying the comprehensive information industry requirements of low energy consumption, speed, bandwidth, density and cost, integrated silicon photonics[2–5] have made rapid progress in a wide range of functionalities such as modulators[6,7], photodetectors[8–10] and optical switches[11]. This is largely enabled by silicon's low loss, low cost and high fabrication compatibility with complementary metal-oxide semiconductor (CMOS) technology[12,13]. One of the most important properties of integrated devices in versatile and reconfigurable photonic networks is highly energy-efficient tunability with a fast response time. Owing to the relatively high thermo-optic coefficient of silicon ($\sim 1.8 \times 10^{-4} \, \mathrm{K}^{-1}$)[14], thermal tuning is often applied using a metallic microheater on a silicon waveguide in tunable silicon micro-ring resonators[15] or Mach-Zehnder interferometers[16] (MZIs). However, to avoid the light-absorption loss induced by the metal, a thick silicon dioxide (SiO$_2$) layer is typically introduced between the silicon waveguide and the metallic heater, inevitably impeding heat transport and dissipation owing to the low thermal conductivity of SiO$_2$ ($1.44 \, \mathrm{W \, m^{-1} \, K^{-1}}$)[17]. Several methods, such as the use of free-standing waveguide structures[18,19] and different doping levels of the waveguide[20], have been proposed to simultaneously obtain lower power consumption and a faster response time. However, free-standing waveguides may lack mechanical stability, and the tuning efficiency of different silicon waveguide doping is as low as $0.12 \, \mathrm{nm \, mW^{-1}}$ (ref. 20).

The use of slow light in silicon photonic crystal waveguides (PhCWs) offers an approach for significantly improving the inherently weak light–matter interaction on nanometer-scale chips. By decreasing the group velocity of the transmitted light in periodic media[21–23], slow light has been utilized in various applications such as sensors[24], amplifiers[25] and nonlinear optics[26]. Meanwhile, owing to many unique properties, such as a zero-band gap and tunable Fermi level[27,28], high carrier mobility[29,30] and ultra-broad absorption bandwidth[31], graphene has been widely merged with nanophotonic structures to enhance the light–matter interaction[32–35]. In addition to these broadly studied applications, the use of graphene as a heating material[36–38] in close contact to the silicon waveguide can significantly improve the tuning efficiency due to graphene's low optical absorption rate[39]. Moreover, owing to the extremely high thermal conductivity of up to $5,300 \, \mathrm{W \, m^{-1} \, K^{-1}}$, response times can be greatly reduced relative to devices with thick SiO$_2$ cladding between the metallic heater and the waveguides. However, the current performances of devices using graphene heaters are limited either by their relatively high power consumptions[37] or by their microsecond response times[38].

In this study, we propose and demonstrate a new concept of enhancing the heater efficiency by the use of slow light in a PhCW with an added layer of graphene working as a heater. Here, graphene can efficiently heat the silicon slow-light photonic crystal, without otherwise perturbing the waveguide and thus jeopardizing the optical performance of the waveguide itself. The active tuning of group velocity in the PhCW has been studied before[23], and here we explore how the slow-light effect in the PhCW can reduce the power consumption. The low optical loss in graphene gives us freedom to optimize the shape of the graphene heater in order to maximize the tuning efficiency, which is significantly different from previous work. We systemically investigate the influence of the graphene–PhCW interaction length and the shape of the graphene heaters on the tuning efficiency. The proposed slow-light-enhanced graphene microheaters show promising potential for applications

in integrated silicon building blocks such as tunable phase shifters and filters that demand low power consumption, a fast response time, and CMOS-compatible fabrication processes.

## Results

**Design of the slow-light-enhanced graphene microheater.** A schematic of the slow-light-enhanced graphene microheater is shown in Fig. 1a. A graphene monolayer is deposited onto the core region of the silicon PhCW. The graphene is contacted by two gold/titanium (Au/Ti) pads, exploiting the low contact resistance between Ti and graphene[40]. Ohmic heating is generated in the graphene via an applied voltage bias between the Au/Ti pads. The width of the graphene overlapping the photonic crystal line defect is designed to be narrower than the other part of the graphene to locally increase the Ohmic dissipation. This results in more effective heating, which will be discussed in the Discussion section.

The slow light can enhance the tuning efficiency owing to the large group index that can be obtained in the PhCW, which increases the effective interaction length between the heater and the waveguides[22], thus increasing the corresponding phase shift. With the aid of perturbation theory[41], the phase shift $\Delta\varphi$ induced by the heating can be expressed as (Supplementary Note 6),

$$\Delta\varphi = \left(\frac{\omega}{c}\right) a \Delta T f L \left(\frac{n_\mathrm{g}}{n_\mathrm{Si}}\right) \qquad (1)$$

where $\omega$ is the angular frequency of the light, $n_\mathrm{g}$ is the group index of the PhCW, $f \equiv \frac{\langle E|\varepsilon|E\rangle_\mathrm{d}}{\langle E|\varepsilon|E\rangle_\mathrm{a}}$ is the filling fraction defining the fraction of the optical field confined in silicon, $L$ is the graphene–PhCW overlap length, $a$ is the thermo-optic coefficient, $\Delta T$ is the temperature increase caused by the graphene heater and $n_\mathrm{Si}$ is the refractive index of silicon, respectively. equation (1) shows how the phase shift (or heating efficiency) benefits from the slow light. The heating efficiency is proportional to the temperature increase ($\Delta T$), which can be optimized by patterning the graphene heater. A longer PhCW can also reduce the power consumption but will increase the insertion loss and the size of the device. The high group index (larger than $\sim 40$) of the PhCW can only be achieved near the fundamental mode cutoff in a very narrow bandwidth where large insertion losses[21] are typical because slow-light promotes further damping[42]. Therefore, the band structure of the PhCW employed here should be carefully optimized to reach a relatively high group index, large bandwidth and low loss in the wavelength regime of interest. These requirements can be satisfied simultaneously using semi-slow-light PhCWs[43,44].

The dispersion relationship of the PhCW is analysed using the plane-wave-expansion method[45]. To obtain a high group index with a large bandwidth, the positions of the first and second rows of holes adjacent to the line-defect are slightly tuned in a W1-type PhCW. The dispersion relationship of the PhCW is presented as the red line in Fig. 1b. We only focus on the even symmetry PhCW mode. After the optimization of the structural parameters with respect to the high group index for a large bandwidth, the lattice constant of 390 nm and the diameter of the holes of 193 nm are chosen, respectively, in our final design. The position of the first row of holes adjacent to the PhCW core is moved 41 nm outward from the original position, and the second row is moved 10 nm outward. Figure 1c displays the calculated (blue line) and measured (green dot) group indices as well as the measured transmission spectrum (red line) (see Supplementary Fig. 1 and Supplementary Note 1). Figure 1d shows a scanning electron microscope image of the fabricated silicon PhCW, where coupling regions are introduced between the strip waveguide and the slow-light PhCW to reduce the coupling loss[46]. The length of

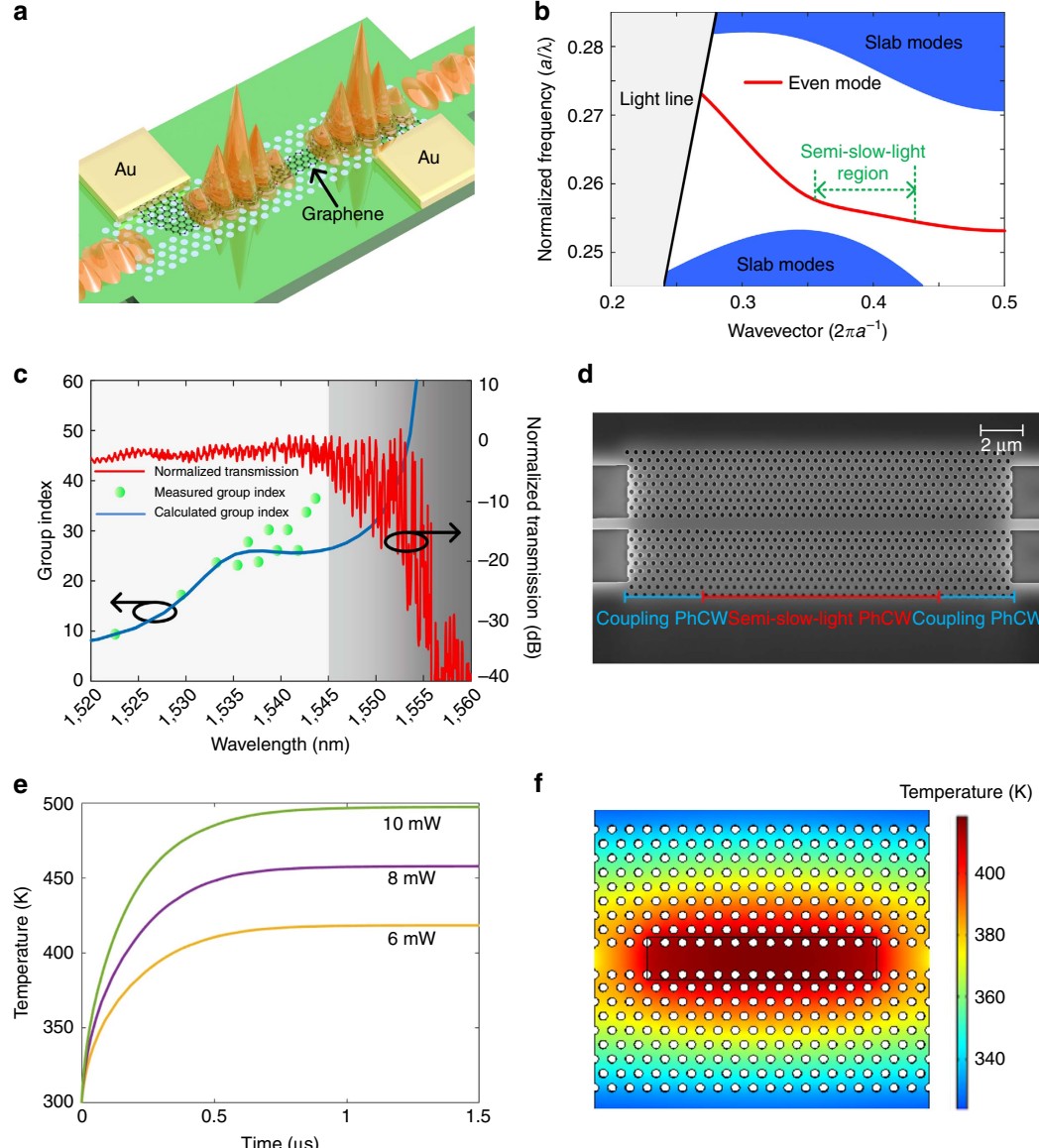

**Figure 1 | Design and theoretical calculation of a slow-light-enhanced graphene microheater.** (**a**) Schematic of the slow-light-enhanced graphene heater. (**b**) Band structure of the photonic crystal waveguide. The even guiding mode (red curve) consists of a semi-slow-light region (green dashed line). The blue-shaded area indicates the slab modes. (**c**) Calculated group index (blue curve), measured group index (green dots) and the transmission spectrum (red curve) of the photonic crystal waveguide. The grey-shaded area indicates the bandgap area of the PhCW. (**d**) The scanning electron microscope image of the fabricated photonic crystal waveguide. (**e**) The temperature response for different heating power. (**f**) The temperature distribution of a graphene–PhCW structure.

the fabricated PhCW is about 10 µm. As shown by the green curve, the fabricated silicon PhCW has a semi-slow-light region at approximately 1,540 nm with a group index of approximately 25; a larger group index can be obtained at longer wavelengths at the cost of larger insertion losses and pronounced fluctuations with wavelength, as shown in the grey area in Fig. 1c. The slight deviation from the theoretical calculation is attributed to the deviations in the fabrication procedure as well as the presence of the thin aluminium oxide layer deposited on the silicon that leads to the tiny shift of the group index curve from the ideal curve to the curve with the longer wavelength range. The trade-off for the slow-light enhancement is a reduced spectral operation bandwidth, which is optimized to approximately 10 nm in our design. Figure 1e shows the temperature response for different heating power, and Fig. 1f illustrates the

temperature distribution of a graphene–PhCW structure. The thermal field is tightly localized in the central area of the PhCW, thus ensuring an efficient heating. Besides, the theoretical response time of the proposed microheater is about 420 ns, which is faster than most previous reported microheaters.

**Device fabrication and characterization.** To characterize the performance of the graphene microheater, we fabricated a tunable MZI filter on a silicon-on-isolator wafer with a 250-nm-thick top silicon layer and a 2 µm $SiO_2$ buried oxide (BOX) layer (see detailed fabrication process in Supplementary Fig. 2 and Supplementary Note 2), as shown in Fig. 2a,b. The transverse electric (TE)-polarized light is coupled from a fibre to the waveguide using a photonic crystal grating coupler[47]. The

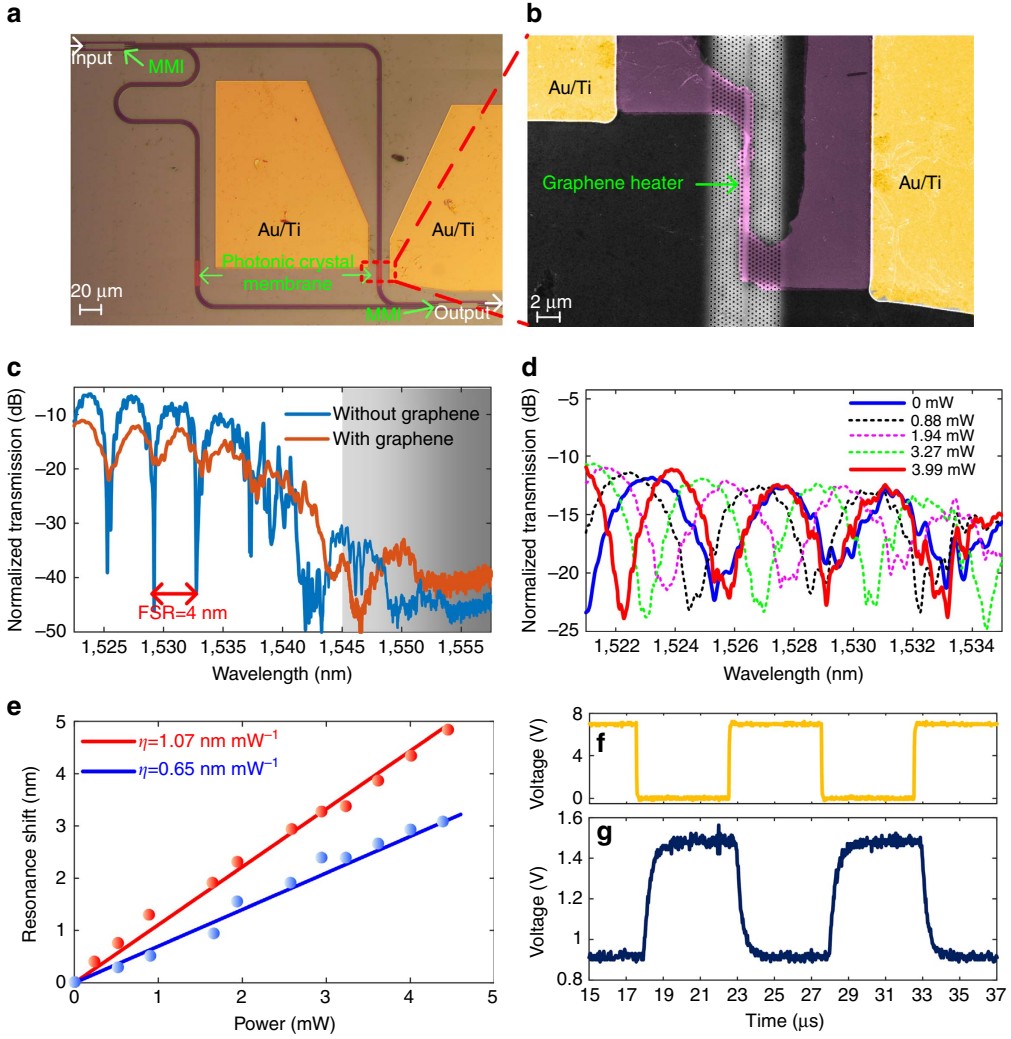

**Figure 2 | Characterization of the slow-light-enhanced graphene microheater.** (**a**) Microscope image of the entire MZI device. (**b**) False-colour scanning electron microscope image of the slow-light-enhanced graphene heater. (**c**) Measured and normalized transmission spectra for the MZI with (red line) and without (blue line) graphene. The grey-shaded area indicates the bandgap area of the PhCW. (**d**) Static response of the heating power. (**e**) Measured resonance shifts for the interference dips at 1525.12 nm (blue) and 1533.71 nm (red) as functions of the applied heating power. (**f**) Driving electrical signal and (**g**) corresponding temporal response signal.

experimental setup is described in detail in Supplementary Fig. 3 and Supplementary Note 3. A multi-mode interferometer divides the input light equally into the two arms of the MZI, both consisting of silicon strip waveguides with dimensions of 450 nm × 250 nm and 28-μm-long PhCW, which includes the slow light region and two coupling regions. The length of graphene–PhCW covering the slow-light waveguide is 20 μm. Raman spectroscopy is employed to examine the quality of graphene after the wet-transfer process, as shown in Supplementary Fig. 5 and Supplementary Note 5, indicating good wet-transfer quality with minor defects. One of the strip waveguides is designed to be 145.7 μm longer than the other in order to induce an appropriate optical path difference. Moreover, to balance the loss induced by the presence of graphene, the PhCW in both arms have equal lengths of graphene on top. Based on the resonant condition of the MZI, we can obtain the resonance shift $\Delta\lambda$ versus the phase shift as $\Delta\lambda = \frac{\Delta\varphi\lambda_0^2}{2\pi n_{si}\Delta L}$, where $\lambda_0$ is the resonance wavelength before heating and $\Delta L$ is the length difference between the two arms of the MZI. In combination with equation (1), we anticipate that the larger group index can induce larger resonance shift.

The transmission spectrum (blue line) of the fabricated MZI without the graphene heater is depicted in Fig. 2c. In our analyses, the measured optical power output is normalized to the reference strip waveguide to exclude the coupling loss of the grating couplers. According to Fig. 2d, the interference dip at 1533.71 nm lies in the slow-light region. Although accompanying lobes and multiple dips, the interference dip can be approximately determined by fitting the measured data with sine functions. The measured free spectral range (FSR) is approximately 4 nm. For comparison, the transmission spectrum of the device incorporating the graphene heater is measured as well, which is shown in the red curve in Fig. 2c. An excess loss of 5 dB is induced, and the extinction ratio is degraded to approximately 8 dB in the MZI with graphene. Except for the loss of 1.1 dB induced by graphene, the excess loss is mainly attributed to metallic contamination during the wet-transfer and lift-off process[48]. Such degradations can be optimized by improving the wet-transfer process by using a modified Radio Corporation of America (RCA) clean process[49].

Next, an external voltage is applied to the graphene heater, and the spectral responses are measured for different applied

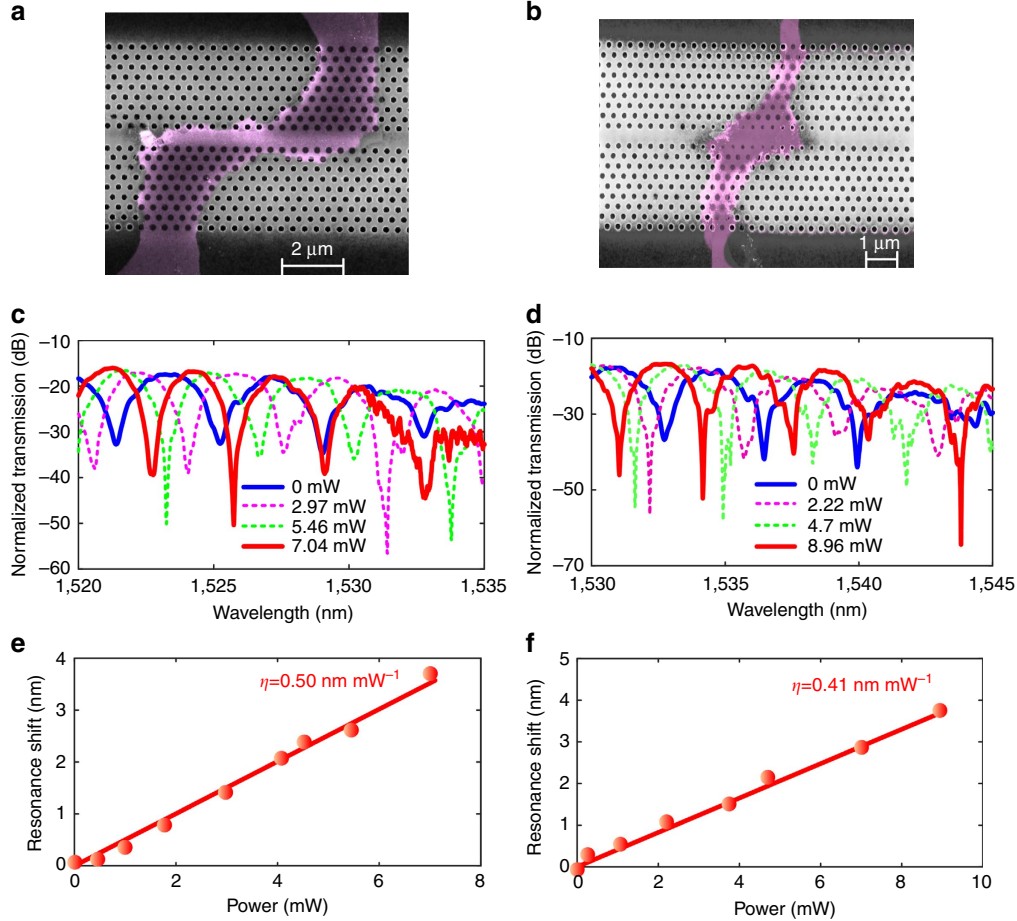

**Figure 3 | Characterization of graphene microheater with different graphene–PhCW interaction lengths.** (**a,b**) The false-colour scanning electron microscope images of the devices with ∼5 and ∼3 μm graphene–PhCW interaction lengths, respectively. Transmission spectra at different heating powers for devices with graphene–PhCW interaction lengths of (**c**) ∼5 μm and (**d**) ∼3 μm. (**e,f**) Corresponding shifts of the transmission spectra as a function of the heating power.

powers (see experimental setup in Supplementary Fig. 3 and Supplementary Note 3); the results are presented in Fig. 2d. The shift of the interference dip at approximately 1533.71 nm reaches one FSR (from the solid blue line to the solid red line) with a tuning power of only 3.99 mW. Meanwhile, there is a larger shift of the interference dip at 1533.71 nm than that for the dip at 1525.12 nm, which agrees well with the theoretical prediction of equation (1), i.e., larger group index (at a longer wavelength) enables higher heating efficiency. Figure 2e plots the shifts of the interference dips at 1533.71 nm (red line) and 1525.12 nm (blue line) as functions of the tuning power, and tuning efficiencies ($\eta$) of 1.07 and 0.65 nm mW$^{-1}$ are achieved, respectively.

The response time of the graphene heater is further characterized by driving the graphene heater with a square waveform electrical signal while the wavelength of the input signal is fixed at 1531.8 nm. The frequency of the driving signal is set to 100 kHz, and the peak-to-peak voltage ($V_{pp}$) is set to 7 V, as shown in Fig. 2f. The $V_{pp}$ of 7 V corresponds to an applied heating power of 3.27 mW. The modulated light is received by a photodetector and recorded by an oscilloscope; the results are shown in Fig. 2g. The 10–90% rising and decaying times are measured to be 750 and 525 ns, respectively. To the best of our knowledge, this is the fastest response time reported for a microheater in silicon photonics. The fast response time is attributed to the silicon photonic crystal membrane with

high thermal conductivity of silicon. The figure of merit, i.e., the product of the power consumption per FSR and the average response time[50], can be used for a comprehensive evaluation of the microheater performance. Our slow-light-enhanced graphene heater has a figure of merit as good as 2.543 nW s, which is one order of magnitude better than previous demonstrations[50].

**The influence of the graphene–PhCW interaction length.** To investigate the impact of the graphene–PhCW interaction length on the tuning efficiency of the heater, we fabricated and measured the MZI with graphene–PhCW overlap lengths of ∼5 and ∼3 μm, shown as in Fig. 3a,b. The measured transmission spectra at different heating powers for graphene–PhCW overlap lengths of ∼5 and ∼3 μm are shown in Fig. 3c,d, respectively, and the corresponding shifts of the transmission spectrum dips as functions of the heating power are depicted in Fig. 3e,f. The resonance shifts are measured at 1532.5 and 1539.2 nm for the case of 5 and 3 μm, respectively. The tuning efficiency decreases when having a shorter graphene–PhCW overlap length, which is in accordance with equation (1) and the thermal distribution mentioned in Supplementary Fig. 4 and Supplementary Note 4. Besides, the interference dips at the shorter wavelength also experience smaller shift under the same heating power, therefore confirming the enhancement due to slow light in both cases.

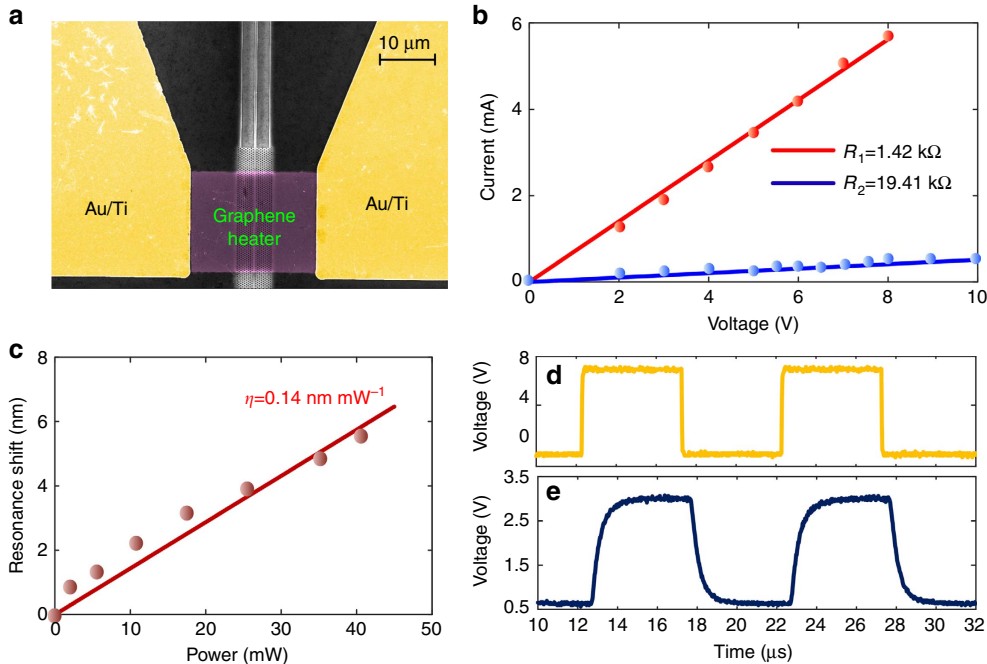

**Figure 4 | Characterization of the straight-shaped graphene microheater.** (**a**) False-colour scanning electron microscope image of the straight-shaped graphene heater. (**b**) Voltage–current relationship for the (blue line) Z-shaped and (red line) straight-shaped heaters. (**c**) Shift of the transmission spectrum as a function of the applied heating power for the straight-shaped heater. (**d**) Driving electrical signal and (**e**) corresponding modulated output signal of the device.

**The influence of the shape of the graphene microheater.** The shape of the graphene heater can have a significant influence on the tuning efficiency. According to our calculated temperature distribution for different shapes of the graphene heater, the tuning efficiency for the narrow-width long-length graphene heater is the highest. The details of the theoretical analysis can be found in Supplementary Fig. 4 and Supplementary Note 4. As shown in Fig. 2b, in the previous design, the graphene coverage on the photonic crystal is designed to be Z-shaped to boost the tuning efficiency. For comparison, a straight-shaped graphene heater fully covering the PhCW membrane is fabricated, as shown in Fig. 4a. Figure 4b shows the voltage–current relationships of both the straight-shaped (red line) and Z-shaped (blue line) graphene layers. The total resistance of the straight-shaped heater is 1.42 kΩ, which is more than 10 times lower than that of the Z-shaped heater. According to the data in Fig. 4c, the heating efficiency is only 0.14 nm mW$^{-1}$ for the straight-shaped heater, much lower than that for the Z-shaped heater. This is because in the Z-shaped case, heat is predominantly generated in the region overlapping the photonic crystal membrane where the light is mainly confined. Thus, the heating-induced modulation of the waveguide can be much more efficient. In contrast, in the straight-shaped case, heat is generated uniformly throughout the entire graphene layer, and only a very small part of the heater effectively overlaps the optical mode; hence, the tuning efficiency is relatively modest, which agrees well with our analytical results. We also test the dynamic response of the device with the straight-shaped heater using the same square waveform electrical signal shown in Fig. 4d and the modulated signal shown in Fig. 4e. Here, we find the rise and decay times to be 850 and 875 ns, respectively. The slower rising time is due to the low heating efficiency, and the slower decay time may be because more overall heat is generated with the same external voltage, requiring more time for dissipation of the heat through the silicon membrane.

## Discussion

In conclusion, we have experimentally demonstrated slow-light-enhanced energy-efficient graphene microheaters with ultrafast response times. We have comprehensively studied the influences of the graphene–PhCW interaction length and the shape of the graphene heater to the heat efficiency. Owing to the slow-light effect, the heating efficiency is as high as 1.04 nm mW$^{-1}$, and the power consumption is as low as 3.99 mW. Furthermore, the 10–90% rising and decaying times are measured to be only 750 and 525 ns, which are the fastest ever reported for microheaters in silicon photonics. To the best of our knowledge, the figure of merit of the proposed device is lower than all previously demonstrated filters based on microheaters. Owing to the widely required applications of microheaters in the photonic integrated circuits, the proposed concept of combining graphene microheater with PhCW provides a promising solution to reduce power consumption in many functional photonic devices, such as optical phase-array antenna, on-chip arbitrary waveform generators, optical switches, phase shifters, tunable filters and modulators. It should be noted that as one of the main challenges in its future application, the relatively high insertion loss could be greatly improved by optimizing the fabrication process to eliminate the metallic contamination during the wet-transfer process[51,52]. Besides, although employing longer PhCW–graphene interaction length may further reduce power consumption, the uniformity in the fabrication of long PhCW should also be considered as another potential challenge as well as the degradation of the graphene microheater. The CMOS-compatible fabrication process of the proposed device enables its wafer-level integration with other nanophotonic devices. The slow-light-enhanced energy-efficient graphene microheater has demonstrated its distinctive advantages in simultaneously achieving both a low power consumption and a fast response time, showing great potential for its use in configurable photonic integrated circuits.

## Methods

**Device fabrication.** The proposed tunable MZI filter was fabricated on an silicon-on-isolator wafer with a 250-nm-thick silicon layer on top of a 2 μm $SiO_2$ buried layer. E-beam lithography and inductively coupled plasma etching were used to fabricate the grating couplers, strip waveguides, multi-mode interferometer and the PhCWs. Standard ultraviolet (UV) lithography was performed to define the wet etch regions used to undercut the PhCWs with AZ5124E acting as the mask. Buffered hydrofluoric acid was used to etch the $SiO_2$ buried layer below the PhCWs. After the membranization, an 11 nm aluminium oxide layer was deposited on the device by atomic-layer deposition. A graphene sheet grown by CVD was wet-transferred onto the silicon device. In the wet-transfer process, AZ resist was first spin-coated onto the graphene covered copper foil and dried at 100 °C for 1 min. Next, an AZ/graphene membrane was obtained by etching away the copper foil in an $Fe(NO_3)_3/H_2O$ solution and then transferring onto the silicon chip. Then, the AZ resist was dissolved in acetone, simultaneously cleaning the graphene surface. The graphene shape was defined using standard UV lithography and $O_2$ plasma etching. Finally, Au/Ti contacts were fabricated on the graphene by standard UV lithography, followed by metal deposition and a lift-off process.

**Data availability.** The data that support the findings of this study are available from the corresponding authors upon request.

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

## Acknowledgements

This work is supported by the Danish Council for Independent Research (DFF-1337 − 00152 and DFF-1335-00771) and the National Natural Science Foundation of China (Grant No. 61622502 and 61475052). The Center for Nanostructured Graphene is sponsored by the Danish National Research Foundation, Project DNRF103. S.Y. is sponsored by the China Scholarship Council (CSC) for supporting his work in Denmark.

## Author contributions

S.Y. and Y.D. proposed the slow-light-enhanced tunable silicon MZI with a graphene heater. S.Y., L.H.F. and S.X. performed numerical simulations. S.Y., Y.D. and X.Z. fabricated the graphene − silicon hybrid tunable MZI device. S.Y. and Y.D. performed the measurements. S.Y., Y.D., S.X. and J.D. discussed and analysed the measured data. S.Y. wrote the first draft of the manuscript. All the authors discussed the results and contributed to the writing of the manuscript.

## Additional information

**Competing financial interests:** The authors declare no competing financial interests.

