## [Peer Review File · Nature Communications]

Reviewers' Comments:

Reviewer #1 (Remarks to the Author)

Graphene-based microheaters with the combination of slow-light effect in silicon photonic crystal waveguides are utilized to enhance the heating efficiency and reduce the response time. The manuscript is well written, although it lacks clarity in certain critical points. Only the experimental results of the enhancement of the heating efficiency by utilizing the slow-light effect of the PhC waveguides are shown. There is only theoretical analysis for the slow light PhC waveguide itself. How the slow light combined with the graphene layer enhance the heating efficiency? Is the experiment result agreed well with the theoretical analysis? Any possible method for improvement?

Some more detailed remarks:

- 1) Explain how the geometry of the slow light PhC cavity is selected. Is it an optimized version?
- 2) From the normalized transmission shown in Fig. 1(d), we can find that the transmission is almost the same for the wavelengths ranging from 1520 nm to 1540 nm. However, we can find a strong decay in the normalized transmission spectrum shown in Fig. 2(c). A transmission difference larger than 15dB can be observed over the 20 nm bandwidth. Can the authors give some explanation for it?
- 3) For the spectra shown in Fig. 2(d), one can find large side lobes/multiple dips in some response curves. How the authors determine the interference dips?
- 4) The authors observe a 5dB excess loss for the MZIs with graphene and they stated that the loss is attributed to metallic contamination during the wet-transfer and lift-off process. Any theoretical loss introduced by the graphene layer itself? Numerical calculations are preferred to verify the author's statement.
- 5) The authors also investigate how the length and shape of the graphene layer influence the efficiency of heating. However, only experimental results are given. It's better to give a detailed analysis by calculate the temperature distributions for different structures. A comparison of the temperature distribution between the one with/without graphene layer will also make a clearer verification for the graphene layer.
- 6) The length of the PhC waveguide is chosen to be 20 μm , what's the length for the graphene layer?

Reviewer #2 (Remarks to the Author)

This manuscript describes high performance graphene micro-heater using slow light structure. The authors report design of slow-light-enhanced graphene microheater and using slow light effect in silicon based photonic crystal structures, it overcome the limitation of high power consumption and response time shown in conventional heaters. Most importantly, they create the fastest microheater in silicon photonics. Also the current manuscript provides guidelines for future microheaters.

The authors' device shows better power efficiency and response time than other groups' devices - graphene heater for silicon photonic device (Schall et al, Optics Express 24 2016) and conventional CMOS-compatible metallic heaters (Masood et al, IEEE 2012). However there is a similar concept and works on photonic crystal waveguides with a micro-heater.

The novelty of the current manuscript looks insufficient for the Nat. Comm.

- Vlasov et al, Nature 2005, vol 438 (this article should be cited)
- Yu et al, Appl. Phys. Lett 2014, 105.
- Ishikura et al, Appl. Phys. Lett. 2012, 100
- Beggs et al, IEEE photonics Tech. Lett. 2009, vol 21

The authors should address that their work goes clearly beyond the earlier works (what is new and novel) and new insights into micro-heater on photonic crystal waveguides.

It can be made suitable for publication after the authors respond to significant benefits over the previous other photonic crystal waveguides heaters that have been explored, as well as address the comments and questions below.

[1] In page 4, The inset in Fig. 2(a) illustrate~ : there is a typo, I feel Fig. 1(a) is correct....

[2] In Fig.1(a), Can the authors make a plot of the temperature profile vs. position and T(K) vs. time(s)?

[2] In Fig.2(g) and Fig.4(e), can you plot the output optical or power signal vs. time (us) instead of voltage(v)?

[3] The authors should add more future potentials and applications as well as challenges in graphene-PhCW

[4] The authors used a CVD-grown graphene. Basically the CVD-grown graphene might have surface defects and copper residues during wet-transfer process. The authors should give more information on their graphene transferred on photonics crystal devices, i.e. Raman analysis, TEM images...

[5] Do the authors see the degradation of graphene heater during heating? Graphene resistors is possibly to experience oxidation coming from defect sites and unprotected graphene is easily deteriorated with absorption of gas molecules, O₂, H₂O from environment.

Reviewer #3 (Remarks to the Author)

The manuscript by Yan et al. proposes a novel method to significantly enhance the thermal tuning efficiency of the microheater in photonic integrated circuits. In their work, monolayer graphene replaces metal to act as a transparent microheater. More importantly, they utilize the slow light effect in photonic crystal waveguide to obtain the high tuning efficiency. They also investigated the influence of the shape of the graphene heater on the device performance, which is valuable. Considering a large number of microheaters or microheater array are required in many functional photonic integrated circuits, such as optical phase array antenna, and on-chip arbitrary waveform generator, the concept of using slow light and graphene to enhance the performance of the microheater is quite attractive. It is meaningful to reduce the power consumption and response time of all microheaters and this manuscript manifests a promising solution.

Overall, the paper is clearly written and organized. And the experiment results are well analyzed and self-consistent. This basic idea of an energy efficient microheater for configurable silicon photonic circuits will be of great interest to the readership of Nature Communications. I can recommend it for publication in this journal providing the following minor issues are fully addressed.

1. In the first paragraph of the Results section, the author wrote: "The inset in Fig. 2(a) illustrates the temperature distribution of the cross section of the proposed microheater when the graphene is powered up, indicating a tight localization of the thermal fields in the silicon membrane." It should be Fig. 1(a) rather than Fig. 2(a). Besides, the thermal field seems not symmetric in the vertical direction, why is that?

2. Why an Al₂O₃ layer is deposited between the waveguide and the graphene heater rather than directly contact the heater on the waveguide?

3. In the fabricated MZI with graphene heater, what is the purpose of making one arm 145.7 μm longer than the other one? How does the author calculate this value? Besides, even if the two arms are in the same length, when one arm is heated by the graphene heater, one can still observe the interference dips at the output.
4. What is the coupling loss of the grating coupler? The authors should illustrate the relationship between the phase shift induced by heating and the shift of resonance dip more clearly.
5. According to Eq. (1), the power consumption should be inversely proportional to the group index. However, the measured results are different from that in the model. Please explain this. Does it mean that there are some other factors limiting the enhancement of the slow light?
6. In the Ref. 37 which also reports graphene microheater, the response time is 12.8 μs and 8.8 μs . In this work, the author claims that the response time is 750ns and 525ns. Does slow light also reduce the response time?
7. What is the position of the resonance dips when the tuning efficiencies (Fig. 3(c) and (d)) are measured in the case of 5 μm and 3 μm graphene heater? Are the heating efficiencies enhanced by slow light in these cases? The authors should clarify that.

Reviewer #1 (Remarks to the Author):

Graphene-based microheaters with the combination of slow-light effect in silicon photonic crystal waveguides are utilized to enhance the heating efficiency and reduce the response time. The manuscript is well written, although it lacks clarity in certain critical points. Only the experimental results of the enhancement of the heating efficiency by utilizing the slow-light effect of the PhC waveguides are shown. There is only theoretical analysis for the slow light PhC waveguide itself. How the slow light combined with the graphene layer enhance the heating efficiency? Is the experiment result agreed well with the theoretical analysis? Any possible method for improvement?

Reply: We thank the Reviewer for bringing the theoretical analysis of our devices to discussion. With the aid of perturbation theory, we now substantiate the original expression for the thermally induced phase shift, when we could obtain that,

$$\Delta\varphi = \left(\frac{\omega}{c}\right) * a * \Delta T * f * L * \left(\frac{n_g}{n_{Si}}\right)$$

where ω is the angular frequency of the light, n_g is the group index of the photonic crystal waveguide, $f \equiv \frac{\langle E|\varepsilon|E \rangle_d}{\langle E|\varepsilon|E \rangle_a}$ is the filling fraction defining the percentage of the optical field in silicon, L is the graphene-PhCW overlap length, a is the thermo-optic coefficient, ΔT is the temperature increase caused by the graphene heater and n_{Si} is the refractive index of silicon, respectively.

From the expression, we can learn that how the phase shift (or heating efficiency) can benefit from the slow light. The heating efficiency is also proportional to the temperature increase (ΔT), which can be optimized by designing the shape of graphene heater. In turn, the temperature raise (ΔT) is proportional to the square of the bias voltage (ΔV^2), i.e. the dissipated power due to Ohmic resistance of the graphene layer. The expression clearly shows how slow light serves to enhance the phase shift caused by heating in the graphene layer that in turn heats the silicon photonic crystal. Our experimental result shown in Fig. 2(e) clearly illustrates the slow-light effect, i.e., the larger group index, the higher heating efficiency. We also presented experimental results of heating efficiency versus the shape of the graphene heater. All experimental results are in good agreement with predictions based on the expression mentioned above. In the revised version, we have updated the Equation 1, and added some discussions when addressing the expression.

The expression gives us a clear way to improve heating efficiency. One of the most possible obvious methods for the improvement is to employ longer PhCW-graphene interaction length. However, this method faces certain challenges. Longer PhCW-graphene interaction length requires higher fabrication uniformity, which may induce larger insertion loss. Another possibility is to optimize the heater shaper in

order to achieve the maximum value of ΔT in response to the applied bias voltage ΔV . Of course, pursuing a higher-group-index region would further improve the efficiency, however it would become challenging due to the increased noise.

Revisions:

[See **page 4, paragraph 3**]:

“The slow light can enhance the tuning efficiency owing to the large group index that can be obtained in the PhCW, which increases the effective interaction length between the heater and the waveguides²², thus increasing the corresponding phase shift. With the aid of perturbation theory⁴¹, the phase shift $\Delta\phi$ induced by the heating can be expressed as,

$$\Delta\phi = \left(\frac{\omega}{c}\right) * a * \Delta T * f * L * \left(\frac{n_g}{n_{Si}}\right) \quad (1)$$

where ω is the angular frequency of the light, n_g is the group index of the photonic crystal waveguide, $f \equiv \frac{\langle E|\varepsilon|E\rangle_d}{\langle E|\varepsilon|E\rangle_a}$ is the filling fraction defining the fraction of the optical field confined in silicon, L is the graphene-PhCW overlap length, a is the thermo-optic coefficient, ΔT is the temperature increase caused by the graphene heater and n_{Si} is the refractive index of silicon, respectively. The Eq. (1) shows how the phase shift (or heating efficiency) benefits from the slow light. The heating efficiency is proportional to the temperature increase (ΔT), which can be optimized by patterning the graphene heater. A longer PhCW can also reduce the power consumption but will increase the insertion loss and the size of the device.”

[Also see **Supplementary Information, Section 6**]

“To achieve the theoretical model of the slow light enhanced efficiency, we employ the perturbation theory. As the phase shift $\Delta\phi$ can be expressed as below, where Δk is the wavevector shift, L is the length of the PhCW.

$$\Delta\phi = \Delta k * L \quad (1)$$

According to the perturbation theory in photonic crystal waveguide, we could obtain that:

$$\Delta k = \left(\frac{\partial\omega}{\partial k}\right)^{-1} * \Delta\omega \quad (2)$$

$$\Delta\omega = -\frac{\omega \langle E|\Delta\varepsilon|E\rangle}{2 \langle E|\varepsilon|E\rangle} \quad (3)$$

Where $\Delta\omega$ is eigenfrequency shift. Therefore, we could obtain that

$$\Delta k = \frac{\omega}{2} * \frac{\Delta\varepsilon}{n_{Si}^2} * \frac{f}{c} * n_g \quad (4)$$

Note that

$$\Delta\varepsilon \approx 2 * \Delta n * n_{Si} \quad (5)$$

Substitute Eqs. (2-5) to Eq. (1), then we could obtain that,

$$\Delta\phi = \left(\frac{\omega}{c}\right) * a * \Delta T * f * L * \left(\frac{n_g}{n_{Si}}\right) \quad (6)$$

Where a is the thermo-optic coefficient, $f \equiv \frac{\langle E| \epsilon | E \rangle_d}{\langle E| \epsilon | E \rangle_a}$ is the filling fraction defining the fraction of the optical field in silicon.”

Some more detailed remarks:

1) Explain how the geometry of the slow light PhC cavity is selected. Is it an optimized version?

Reply: We thank the Reviewer for posing an important question. The selected geometry of the fabricated photonic crystal has been optimized using 3D-MPB software during our initial design work. Based on the structure in Ref. 44, we refined the structural parameters of the photonic crystal waveguide, including the size of the air holes, the position of holes of the first and second rows, and the thickness of the silicon membrane, which is set as 250 nm in our calculation. Our aim is to obtain relatively high group index as well as large bandwidth of the even mode by optimizing these parameters.

Revision:

[See **page 6, paragraph 1**]:

“After the optimization of the structural parameters with respect to high group index for a large bandwidth, the lattice constant of 390 nm, and the diameter of the holes of 193 nm are chosen respectively in our final design. The position of the first row of holes adjacent to the PhCW core is moved 41 nm outward from the original position, and the second row is moved 10 nm outward.”

2) From the normalized transmission shown in Fig. 1(d), we can find that the transmission is almost the same for the wavelengths ranging from 1520 nm to 1540 nm. However, we can find a strong decay in the normalized transmission spectrum shown in Fig. 2(c). A transmission difference larger than 15dB can be observed over the 20 nm bandwidth. Can the authors give some explanation for it?

Reply: We thank the Reviewer for raising this comment. We believe the difference in transmission spectrum between Fig. 1(d) (Fig. 1(c) in the revised manuscript) and Fig. 2(c) comes from the fabrication process in our nano-fabrication platform. On one hand, in Fig. 1(c), the length of the measured photonic crystal waveguide is 10 μm . On the other hand, in Fig. 2(c), the length of the photonic crystal waveguide in the MZI is 28 μm . According to our previous fabrication results, the transmission of longer photonic crystal waveguide is rather sensitive to the uniformity of the holes since the longer length requires larger E-beam exposure area. This increased sensitivity to fabrication disorder is an inherent challenge of slow-light photonic crystals. As a result, the longer photonic crystal waveguide may experience significant larger insertion loss as well as the transmission spectral shift towards shorter wavelength, compared with shorter waveguide with the presumed same structural parameters.

Fig. R1. Transmission spectra as a function of wavelength of PhCW with different length.

This phenomenon has been observed in our previously fabricated photonic crystal waveguides, as shown in Fig. R1. In the figure above, the red, blue and purple curves represent the measured transmission spectra of the nominally same photonic crystal design with the length of 20 μm , 50 μm and 100 μm , respectively. We can see that the transmission spectrum of the longer photonic crystal waveguide has a much larger insertion loss than the shorter PhCW. Therefore, we may conclude that the problem reviewer mentioned mainly results from the fabrication uniformity degradation caused by the longer PhCW. Besides, the measured transmission spectrum in Fig. 2(c) also includes the insertion loss of two MMIs in the MZI, which can further increase the loss in Fig. 2(c).

Furthermore, we have used the opportunity to clarify the length of the measured photonic crystal waveguide of Fig. 1(c) and Fig. (2) in the revised manuscript.

Revision:

[See **Page 6, Paragraph 1**]

“The inset of Figure 1(c) shows a scanning electron microscope (SEM) image of the fabricated silicon PhCW, where coupling regions are introduced between the strip waveguide and the slow-light PhCW to reduce the coupling loss⁴⁵. The length of the fabricated PhCW is about 10 μm .”

[See **Page 7, Paragraph 2**]

*“A multi-mode interferometer (MMI) divides the input light equally into the two arms of the MZI, both consisting of silicon strip waveguides with dimensions of 450 nm*250 nm and 28- μm -long PhCWs, which includes the slow light region and two coupling regions. The length of graphene-PhCW covering the slow-light waveguide is 20 μm .”*

3) For the spectra shown in Fig. 2(d), one can find large side lobes/multiple dips in some response curves. How the authors determine the interference dips?

Reply: We thank the Reviewer for the insightful comments. When large side lobes occurred in the measurement, we used sine functions to fit the measured data since the ideal transfer function of the MZI is in the form of sine. The following figure displays the fitting result of the interference dips with multiple dips in Fig. 2(d). Note that we have converted the log unit of the normalized transmission to linear unit. Therefore, we can approximately determine the resonance dip of the transfer function to be 1533.71 nm in this case. In the revised manuscript, we have added the description of how to determine the resonance dip with multiple dips or large side lobes.

Revision:

[See **page 8, paragraph 1**]

“According to Fig. 2(d), the interference dip at 1533.71 nm lies in the slow-light region. Although accompanying lobes and multiple dips, the interference dip can be approximately determined by fitting the measured data with sine functions.”

4) The authors observe a 5dB excess loss for the MZIs with graphene and they stated that the loss is attributed to metallic contamination during the wet-transfer and lift-off process. Any theoretical loss introduced by the graphene layer itself? Numerical calculations are preferred to verify the author’s statement.

Reply: We thank the Reviewer for bringing our attention to the 5dB excess loss and the loss introduced by the graphene. We have implemented a full-wave simulation (with commercial available Lumerical FDTD solutions) to evaluate the loss by graphene. In the modelling, we use the carrier relaxation time of 50 meV and the Fermi level of 0.3eV for graphene, i.e. parameters close to the measured values for the CVD-grown graphene. We find that the additional loss introduced by graphene is around 1.1 dB, see below, for our fabricated 20 um-length device. Therefore, we are

naturally led to conclude that the 5 dB loss is mainly attributed to metallic contamination during the wet-transfer and lift-off process. In the revised manuscript, we have rephrased our statement as follows.

Revision:

[See page 9, paragraph 1]

“An excess loss of 5 dB is induced, and the extinction ratio is degraded to approximately 8 dB in the MZI with graphene. Except for the loss of 1.1 dB induced by graphene, the excess loss is mainly attributed to metallic contamination during the wet-transfer and lift-off process⁴⁷.”

5) The authors also investigate how the length and shape of the graphene layer influence the efficiency of heating. However, only experimental results are given. It's better to give a detailed analysis by calculate the temperature distributions for different structures. A comparison of the temperature distribution between the one with/without graphene layer will also make a clearer verification for the graphene layer.

Reply: We thank the Reviewer for the constructive suggestion to study the temperature distributions for different structures. We have calculated the temperature distributions with respect to different length and shape of the graphene layer. In order to see the shape effect clearly, we only considered the contributions from the central part when having the same normalized heat power. The generated temperature distribution really indicates that the narrow-width long-length graphene-PhCW gives the largest thermal effect, which agrees well with our experimental results. The temperature dynamics shown in Fig. R2(d) exhibits a fast time response with 420 ns, very close to what we observed in experiments. Here the temperature response is mainly determined by silicon, being roughly independent of the length and shape of graphene, which is also confirmed by our experiments. Since

the graphene is considered as a perfect conductor, we cannot see any difference for the temperature distribution with or without the graphene layer.

In the revised version, we have added the thermal characteristics in Fig. 1(d) and corresponding discussion is included in the Supplementary Information.

Fig. R2. Upper: Temperature distribution of a PhC waveguide covered by graphene with a long coverage length, a short coverage length, and full coverage on the PhC waveguide. Down: The temperature response for different heating power.

Revision:

[See **page 6, paragraph 1**]

“Figure 1(d) shows the temperature response for different heating power, where the inset illustrates the temperature distribution of a graphene-PhCW structure. The thermal field is tightly localized in the central area of the photonic crystal waveguide, thus ensuring an efficient heating. Besides, the theoretical response time of the proposed microheater is about 420 ns, which is faster than most previous reported microheaters.”

[Also see **page 10, paragraph 2**]

“The resonance shifts are measured at 1532.5nm and 1539.2nm for the case of 5 μm

and 3 μm , respectively. The tuning efficiency decreases when having a shorter graphene-PhCW overlap length, which is in accordance with Eq. (1) and the thermal distribution mentioned in the Supplementary Information.”

[Also see **page 11, paragraph 1**]

“The shape of the graphene heater can have a significant influence on the tuning efficiency. According to our calculated temperature distribution for different shapes of the graphene heater, the tuning efficiency for the narrow-width long-length graphene heater is the highest. The details of the theoretical analysis can be found in the Supplementary Information Section 4.”

[Also see **Supplementary Information, Section 4**]

“To study the temperature distributions for different structures, we have calculated the temperature distributions with respect to different length and shape of the graphene layer as follows from the same heating power. By solving the Poisson equation, we developed a 3D numerical model to achieve both temperature distribution and dynamic response. In order to see the shape effect clearly, we only considered the contributions from the central part when having the same normalized heat power. As seen from Fig. S4(a) and (b), the graphene-PhCW with longer length can have a larger temperature increase under the same normalized heating power. Besides, according to Fig. S4(a) and (c), the full coverage graphene heater on the photonic crystal waveguide, i.e. “Straight-shaped” (Fig. S4(c)), results in a much lower temperature increase compared to the structure covering the central part of photonic crystal waveguide (Fig. S4(a)). Therefore, the tuning efficiency of the narrow-width long-length graphene heater can provide high heating efficiency under the same normalized power.”

6) The length of the PhC waveguide is chosen to be 20 μm , what’s the length for the graphene layer?

Reply: We thank the Reviewer for directing us to our ambiguous description. Here, the length of the graphene-PhCW is 20 μm , while the total length of the PhCW is 28 μm , including 4 μm -long coupling region on both sides, as shown in Fig. 1(c). We have corrected this unclear description in our revised manuscript.

Revision:

[See **page 7, paragraph 2**]

“A multi-mode interferometer (MMI) divides the input light equally into the two arms of the MZI, both consisting of silicon strip waveguides with dimensions of 450 nm*250 nm and 28- μm -long PhCW, which includes the slow light region and two coupling regions. The length of graphene-PhCW covering the slow-light waveguide is 20 μm .”

Reviewer #2 (Remarks to the Author):

This manuscript describes high performance graphene micro-heater using slow light structure. The authors report design of slow-light-enhanced graphene microheater and using slow light effect in silicon based photonic crystal structures, it overcome the limitation of high power consumption and response time shown in conventional heaters. Most importantly, they create the fastest microheater in silicon photonics. Also the current manuscript provides guidelines for future microheaters.

The authors' device shows better power efficiency and response time than other groups' devices - graphene heater for silicon photonic device (Schall et al, Optics Express 24 2016) and conventional CMOS-compatible metallic heaters (Masood et al, IEEE 2012). However there is a similar concept and works on photonic crystal waveguides with a micro-heater.

The novelty of the current manuscript looks insufficient for the Nat. Comm.

- Vlasov et al, Nature 2005, vol 438 (this article should be cited)
- Yu et al, Appl. Phys. Lett 2014, 105.
- Ishikura et al, Appl. Phys. Lett. 2012, 100
- Beggs et al, IEEE photonics Tech. Lett. 2009, vol 21

The authors should address that their work goes clearly beyond the earlier works (what is new and novel) and new insights into micro-heater on photonic crystal waveguides.

Reply: We sincerely thank the Reviewer for bringing these published papers to discussion. These four papers are all related to photonic crystal waveguides, however, they use a conventional metal-heater or lateral electrical contact to tune optical properties. In the work mentioned above, authors focused on thermal tuning of group velocity in photonic crystal waveguide (Vlasov *et al.*, Nature 2005), group delay or dispersion (Ishikura *et al.*, Appl. Phys. Lett. 2012). Beggs *et al.* employed the photonic crystal waveguide directional coupler with metallic heater (IEEE Photonics Tech. Lett. 2009) to realize optical switch with 20 μ s response time.

The main contribution and novelty of our work is to demonstrate a new concept of enhancing heater efficiency by use of slow light in photonic crystal waveguide with the combination of graphene as a heater. Here, graphene can efficiently heat the silicon slow-light photonic crystal, without otherwise perturbing the waveguide and thus not jeopardizing the optical performance of the waveguide itself. This is now small feat and since the graphene is situated in the very near-field of the waveguide this is only possible because the graphene layer is atomically thin (relatively modest light-graphene interactions as opposed to the presence of a much thicker conducting metal layer introducing significant optical losses). Supported by a relatively simple

theory in Eq. (1), we for the first time proposed that the energy consumption for the heater can be reduced by using the slow-light effect arisen in the photonic crystal waveguide. Relying on low-loss graphene, we optimized the physical shape of the graphene heater to achieve the highest efficiency, significantly different from what people did previously.

With this new concept, we demonstrated the graphene-based microheater for silicon photonics with the efficiency of 1.07 nm/mW, much higher than metal-based heaters, and at the same time, the response time of 525 ns is much faster than previous graphene-based heaters. To the best of our knowledge, this is the first work about graphene-based microheaters in silicon photonic-crystal systems, which includes at least four hot research topics, i.e., “graphene”, “slow-light”, “photonic crystal”, and “silicon photonics”. Given the novelty of our work and good performances of our devices, we believe that our work will be of great interest to the readership of Nature Communications, which was also pointed out by the 3rd Reviewer.

According to the Reviewer's comments, we have made the revision to emphasize more on the novelty of our work. The Nature paper by Vlasov has also been added to the reference.

Revision:

[See **page 3, paragraph 2**]

“In this study, we propose and demonstrate a new concept of enhancing the heater efficiency by the use of slow light in a photonic crystal waveguide with an added layer of graphene working as a heater. Here, graphene can efficiently heat the silicon slow-light photonic crystal, without otherwise perturbing the waveguide and thus jeopardizing the optical performance of the waveguide itself. The active tuning of group velocity in the photonic crystal waveguide has been studied before²³, and here we explore how the slow-light effect in the photonic crystal waveguide can reduce the power consumption. The low optical loss in graphene gives us freedom to optimize the shape of the graphene heater in order to maximize the tuning efficiency, which is significantly different from previous work. We systemically investigate the influence of the graphene-PhCW interaction length and the shape of the graphene heaters on the tuning efficiency. The proposed slow-light-enhanced graphene microheaters show promising potential for applications in integrated silicon building blocks such as tunable phase shifters and filters that demand low power consumption, a fast response time, and CMOS-compatible fabrication processes.”

[Also See **References 23**]

23. Vlasov YA, O'Boyle M, Hamann HF, McNab SJ. Active control of slow light on a chip with photonic crystal waveguides. *Nature* 438, 65-69 (2005).

It can be made suitable for publication after the authors respond to significant

benefits over the previous other photonic crystal waveguides heaters that have been explored, as well as address the comments and questions below.

[1] In page 4, The inset in Fig. 2(a) illustrate~ : there is a typo, I feel Fig. 1(a) is correct....

Reply: We thank the Reviewer for pointing out the typo, which we have corrected it in the revised manuscript.

[2] In Fig.1(a), Can the authors make a plot of the temperature profile vs. position and T(K) vs. time(s)?

Reply: We thank the Reviewer for encouraging us to study thermal characteristic, which is also proposed by the other two reviewers. We have made effort to develop a 3D numerical model by solving the Poisson equation, and have achieved both the temperature distribution and the temperature response. The temperature response indicates a fast time response with ~ 420 ns, which is in good agreement with our experimental values. In the revised version, we have made a new subplot in Fig. 1 as follows.

Besides, in the revised version, we have added the thermal characteristics and corresponding discussion in the Supplementary Information.

Revisions:

[See **page 6, paragraph 1**]

“Figure 1(d) shows the temperature response for different heating power, where the inset illustrates the temperature distribution of a graphene-PhCW structure. The thermal field is tightly localized in the central area of the photonic crystal waveguide, thus ensuring an efficient heating. Besides, the theoretical response time of the

proposed microheater is about 420 ns, which is faster than most previous reported microheaters.”

[2] In Fig.2(g) and Fig.4(e), can you plot the output optical or power signal vs. time (us) instead of voltage(v)?

Reply: We thank the Reviewer for drawing our attention to the optical power as a function of time. In our experiment, the modulated optical signal is firstly received by a photodetector and then recorded by an oscilloscope; see the setup in the Supplementary Information. Therefore, we provide the results for the voltage vs. time, similar to previous works [Yu *et al.*, *Optica* **3**, 2 (2016), Gan *et al.*, *Nanoscale* **7**, 47 (2015)]. Since the converted output electrical signal is proportional to the output optical signal power, both the voltage and optical signal should be able to deliver the temporal response of the microheater. We have made some revision in the Supplementary Information to clarify our experimental setup as follows.

Revision:

[See **Supplementary Information, Section 3**]:

“The modulated signal is received by a photodetector, which converts the optical signal to the electrical signal, which is then recorded by an oscilloscope.”

[3] The authors should add more future potentials and applications as well as challenges in graphene-PhCW

Reply: We thank the Reviewer for encouraging us to elaborate more on future potentials, applications and challenges in graphene-PhCW. Following this insightful advice, we have added some discussion in the revised manuscript, see below.

Revision:

[See **Page 13, Paragraph 1**]:

“Due to the widely required applications of microheaters in photonic integrated circuits, the proposed concept of combining graphene microheater with PhCW provides a promising solution to reduce power consumption in many functional photonic devices, such as optical phase-array antenna, on-chip arbitrary waveform generators, optical switches, phase shifters, tunable filters and modulators. It should be noted that as one of the main challenges in its future application, the relatively high insertion loss could be greatly improved by optimizing the fabrication process to eliminate the metallic contamination during the wet-transfer process^{51,52}. Besides, although employing longer PhCW-graphene interaction length may further reduce power consumption, the uniformity in the fabrication of long photonic crystal waveguide should also be considered as another potential challenge as well as the degradation of the graphene microheater.”

[4] The authors used a CVD-grown graphene. Basically the CVD-grown graphene might have surface defects and copper residues during wet-transfer process. The authors should give more information on their graphene transferred on photonics crystal devices, i.e. Raman analysis, TEM images...

Reply: We thank the Reviewer for the valuable suggestion. We have added the Raman spectrum and its corresponding discussion in the supplementary information.

Revision (see the Supplementary Information, section 5):

“Here, we use Raman spectroscopy to examine the quality of graphene after the wet-transfer process. For the pristine CVD graphene, normally there is only a very small Raman D peak, indicative of the good structural quality of graphene. Fig. S5 shows the Raman spectrum of graphene after the wet-transfer process, where we observe a prominent D peak appearing at 1358 cm^{-1} . The observation suggests the presence of defects during the wet-transfer process”.

[5] Do the authors see the degradation of graphene heater during heating? Graphene resistors is possibly to experience oxidation coming from defect sites and unprotected graphene is easily deteriorated with absorption of gas molecules, O₂, H₂O from environment.

Reply: We thank the Reviewer for comments with respect to degradation and oxidation of graphene. We agree with the Reviewer that graphene may experience oxidation in the O₂ atmosphere [see Liu *et al.*, Nano Lett. **8**, 1965 (2008)]. In our experiment, we did not find any degradation of graphene during heating, and we have repeated the measurement several times and found no significant change for the device performance.

Reviewer #3 (Remarks to the Author):

The manuscript by Yan et al. proposes a novel method to significantly enhance the thermal tuning efficiency of the microheater in photonic integrated circuits. In their work, monolayer graphene replaces metal to act as a transparent microheater. More importantly, they utilize the slow light effect in photonic crystal waveguide to obtain the high tuning efficiency. They also investigated the influence of the shape of the graphene heater on the device performance, which is valuable. Considering a large number of microheaters or microheater array are required in many functional photonic integrated circuits, such as optical phase array antenna, and on-chip arbitrary waveform generator, the concept of using slow light and graphene to enhance the performance of the microheater is quite attractive. It is meaningful to reduce the power consumption and response time of all microheaters and this manuscript manifests a promising solution.

Overall, the paper is clearly written and organized. And the experiment results are well analyzed and self-consistent. This basic idea of an energy efficient microheater for configurable silicon photonic circuits will be of great interest to the readership of Nature Communications. I can recommend it for publication in this journal providing the following minor issues are fully addressed.

1. In the first paragraph of the Results section, the author wrote: "The inset in Fig. 2(a) illustrates the temperature distribution of the cross section of the proposed microheater when the graphene is powered up, indicating a tight localization of the thermal fields in the silicon membrane." It should be Fig. 1(a) rather than Fig. 2(a). Besides, the thermal field seems not symmetric in the vertical direction, why is that?

Reply: We thank the Reviewer for the positive recommendation for publication in Nature Communications. We also thank the Reviewer for directing us to the labeling of figures. We have corrected this typo in the revised manuscript.

The asymmetry of the thermal field in the vertical direction comes from the asymmetric surroundings of the photonic crystal membranes. In our experiment, we used the BHF to etch away the silicon dioxide BOX layer. Due to the short etching time, there remains some silicon dioxide BOX layer, leading to the asymmetry of the whole structure in the vertical direction. To better illustrate the temperature distribution and understand the dependence of the tuning efficiency to the graphene shape, we have replotted the temperature distribution, see the inset in Fig. 1(d), and have added the temperature response in Fig. 1(d) in the revised manuscript.

Revision:

[See **page 6, paragraph 1**]

“Figure 1(d) shows the temperature response for different heating power, where the inset illustrates the temperature distribution of a graphene-PhCW structure. The thermal field is tightly localized in the central area of the photonic crystal waveguide, thus ensuring an efficient heating. Besides, the theoretical response time of the proposed microheater is about 420 ns, which is faster than most previous reported microheaters.”

2. Why an Al₂O₃ layer is deposited between the waveguide and the graphene heater rather than directly contact the heater on the waveguide?

Reply: We thank the Reviewer for bringing the Al₂O₃ to discussion. The reason for depositing the Al₂O₃ layer between the waveguide and the graphene is to avoid the direct contact between Au/Ti contact and the silicon. If the metal contact is directly deposited on the silicon waveguide, when we apply the external voltage, there might be current passing through the waveguide, which would then also heat the waveguide. In this case, the tuning efficiency can be affected, and the efficiency of the graphene microheater cannot be unambiguously evaluated. Therefore, we deposit the Al₂O₃ layer between the silicon waveguide and graphene.

Based on the Reviewer’s question, we have added some description in the Supplementary Information regarding the Al₂O₃.

Revision:

[See **Supplementary Information, Section 2**]

“An 11nm Al₂O₃ layer is deposited on the wafer using atomic layer deposition (Picosun ALD model R200), shown in Fig. S2(e), in order to avoid the direct contact between metal contacts and the silicon waveguide as well as to improve the surface smoothness.”

3. In the fabricated MZI with graphene heater, what is the purpose of making one arm 145.7 um longer than the other one? How does the author calculate this value? Besides, even if the two arms are in the same length, when one arm is heated by the graphene heater, one can still observe the interference dips at the output.

Reply: We appreciate the reviewer’s comment. The purpose of making one arm longer than the other one is to provide a comb-like transmission at our desired FSR (4nm in our case). This makes it easier to observe the phase change induced by the graphene microheater, and to evaluate the tuning efficiency by measuring the shift of the interference dips. The length difference (145.7 um) is obtained by

$$\Delta L = \frac{\lambda^2}{FSR \cdot n_g}$$

where the wavelength (λ) is 1550 nm, and the group index (n_g) of the silicon strip waveguide is 4.12.

4. What is the coupling loss of the grating coupler? The authors should illustrate the relationship between the phase shift induced by heating and the shift of resonance dip more clearly.

Reply: Thanks to the reviewer's comment. The coupling loss of the grating coupler is about 6 dB each side. Based on the resonant condition of the MZI, we can obtain the resonance dip shift $\Delta\lambda$ as a function of the phase shift $\Delta\varphi$:

$$\Delta\lambda = \frac{\Delta\varphi\lambda_0^2}{2\pi n_{Si}\Delta L}$$

where λ_0 is the resonance wavelength before heating, λ_1 is the resonance wavelength after heating, n_{Si} is the effective index of the silicon waveguide, and ΔL is the length difference between the two arms of the MZI.

From the equation, we can easily find that larger phase shifts can induce larger resonance shifts. We have revised the manuscript to address the reviewer's concern.

Revision:

[See **page 7, paragraph 2**]

“Based on the resonant condition of the MZI, we can obtain the resonance shift $\Delta\lambda$ versus the phase shift as $\Delta\lambda = \frac{\Delta\varphi\lambda_0^2}{2\pi n_{Si}\Delta L}$, where λ_0 is the resonance wavelength before heating, and ΔL is the length difference between the two arms of the MZI. In combination with Eq. (1), we anticipate that larger group index can induce larger resonance shift.”

[Also see **Supplementary Information, Section 6**]

“To obtain the relationship between phase shift and the resonance dip, resonant condition of the MZI is employed:

$$\frac{2\pi}{\lambda_0} * n_{Si} * \Delta L = (2k + 1)\pi \quad (6)$$

Where λ_0 is the resonance wavelength before heating, n_{Si} is the effective index of the silicon waveguide, ΔL is the length difference between the two arms of the MZI and k is a positive integer. When the phase shift is induced to the one arm of the MZI, we can achieve that,

$$\frac{2\pi}{\lambda_1} * n_{Si} * \Delta L + \Delta\varphi = (2k + 1)\pi \quad (7)$$

Note that,

$$\Delta\varphi = \left(\frac{2\pi}{\lambda_1}\right) * a * \Delta T * f * L * \left(\frac{n_g}{n_{Si}}\right) \quad (8)$$

$$\lambda_0^2 \approx \lambda_0 * \lambda_1 \quad (9)$$

Where λ_1 is the resonance wavelength after heating, n_g is the group index of the PhCW and ΔL is the length difference between the two arms of the MZI.

Combining the Eqs. (6-9), then we can achieve that:

$$\Delta\lambda = \frac{\Delta\varphi\lambda_0^2}{2\pi n_{Si}\Delta L} \quad “$$

5. According to Eq. (1), the power consumption should be inversely proportional to the group index. However, the measured results are different from that in the model. Please explain this. Does it mean that there are some other factors limiting the enhancement of the slow light?

Reply: We thank the Reviewer for raising the discussion of the relation between the power consumption and group index. As shown in Fig. 2(d) and 2(e), higher heating efficiency (namely lower power consumption) is observed for longer wavelength, i.e. for larger group index, which is same as that predicted by Eq. (1). We have clarified this relation in the revised manuscript.

Revision:

[See **page 9, paragraph 1**]

“The shift of the interference dip at approximately 1533.71 nm reaches one FSR (from the solid blue line to the solid red line) with a tuning power of only 3.99 mW. Meanwhile, there is a larger shift of the interference dip at 1533.71 nm than that for the dip at 1525.12 nm, which agrees well with the theoretical prediction of Eq. (1), i.e., larger group index (at a longer wavelength) enables higher heating efficiency”

6. In the Ref. 37 which also reports graphene microheater, the response time is 12.8 us and 8.8 us. In this work, the author claims that the response time is 750 ns and 525 ns. Does slow light also reduce the response time?

Reply: We acknowledge the Reviewer’s comment. In our work, the slow-light effect mainly contributes to the enhanced heating efficiency, according to our model and experimental results. The fast response time is mainly due to the silicon photonic crystal membrane, where the heat dissipates within the plane. Since silicon has a much higher thermal conductivity ($149 \text{ W}\cdot\text{m}^{-1}\cdot\text{K}^{-1}$) than silicon dioxide, we thus observe a fast time response. In the revised version, we have also emphasized the reason for the fast time response.

Revision:

[See **page 10, paragraph 1**]:

“The fast response time is attributed to the silicon photonic crystal membrane with high thermal conductivity of silicon.”

7. What is the position of the resonance dips when the tuning efficiencies (Fig. 3(c) and (d)) are measured in the case of 5 um and 3 um graphene heater? Are the heating efficiencies enhanced by slow light in these cases? The authors should clarify that.

Reply: We thank the Reviewer for the comments. The positions of the resonance dips of 5 μm and 3 μm cases are 1532.5 nm and 1539.2 nm, respectively. The tuning efficiency is improved since we observe the larger resonance shift at the longer

wavelength in both cases of 5 μm and 3 μm . We have added the missing information of the measured wavelength and clarify the slow light enhancement in both cases of 5 μm and 3 μm in the revised manuscript.

Revision:

[See **page 10, paragraph 2**]:

“The resonance shifts are measured at 1532.5 nm and 1539.2 nm for the case of 5 μm and 3 μm , respectively. The tuning efficiency decreases when having a shorter graphene-PhCW overlap length, which is in accordance with Eq. (1) and the thermal distribution mentioned in the Supplementary Information. Besides, the interference dips at the shorter wavelength also experience smaller shift under the same heating power, therefore confirming the enhancement due to slow light in both cases.”

REVIEWERS' COMMENTS:

Reviewer #1 (Remarks to the Author):

The authors have addressed all my concerns. I recommend the publication of the manuscript.

Reviewer #2 (Remarks to the Author):

The answers the authors made on the comments are well addressed. I recommend it for publication.

Reviewer #3 (Remarks to the Author):

Even though other two referees have doubts on some technique issues which actually do not harm the novelty and originality of this work, I feel most of the comments have been sufficiently addressed in the revised draft which has been significantly improved. Anyway, all my comments have been fully addressed in the revised version, and I can recommend this version for publication in Nature Communications.